# Four New Diterpenoids from the South China Sea Soft Coral *Sinularia nanolobata* and DFT-Based Structure Elucidation

**DOI:** 10.3390/molecules28196892

**Published:** 2023-09-30

**Authors:** Dan-Dan Yu, Lin-Mao Ke, Jiao Liu, Song-Wei Li, Ming-Zhi Su, Li-Gong Yao, Hui Luo, Yue-Wei Guo

**Affiliations:** 1College of Pharmacy, Guangdong Medical University, Zhanjiang 524023, China; ddyu@baridd.ac.cn (D.-D.Y.); klm102198@163.com (L.-M.K.); 2State Key Laboratory of Drug Research, Shanghai Institute of Materia Medica, Chinese Academy of Sciences, Shanghai 201203, China; 201728012342081@simm.ac.cn (J.L.); yaoligong@simm.ac.cn (L.-G.Y.); 3School of Medicine, Shanghai University, Shanghai 200444, China; simmswli@163.com; 4Shandong Laboratory of Yantai Drug Discovery, Bohai Rim Advanced Research Institute for Drug Discovery, Yantai 264117, China; smz0310@163.com

**Keywords:** marine natural product, *Sinularia nanolobata*, structure elucidation, cytotoxicity

## Abstract

Three new cembranoids (**1**–**3**) and a new casbanoid (**4**), along with three known analogues (**5**–**7**), have been isolated from the soft coral *Sinularia nanolobata* collected off Ximao Island. The structures, including the absolute configurations of new compounds, were established using extensive spectroscopic data analysis, time-dependent density functional theory/electronic circular dichroism (TDDFT-ECD) calculations, and the comparison with spectroscopic data of known compounds. In the in vitro bioassay, compounds **1** and **5** exhibited moderate cytotoxic activities against human erythroleukemia (HEL) cell lines, with IC_50_ values of 37.1 and 42.4 μM, respectively.

## 1. Introduction

Soft corals of the genus *Sinularia* (subclass Octocorallia, order Alcyonacea, family Alcyoniidae) have been well studied by organic chemists for a long time [1,2]. To date, hundreds of secondary metabolites have been discovered in approximately 50 species of this genus [2]. Chemically, the structures of these metabolites can be classified into three main types, namely terpenes, steroids, and prostaglandins, which are responsible for a diverse range of significant bioactivities, especially cytotoxic and anti-inflammatory potentials [1,2,3].

The first chemical and biological study on the soft coral *S. nanolobata* was performed in 1997, resulting in the isolation of four cytotoxic amphilectane-type diterpenoids [4]. In the following decades, a variety of diterpenoids, nor-diterpenoids, sesquiterpenoids, nor-sesquiterpenoids, steroids, seco-steroids, and steroidal glycosides, which exhibited interesting biological activities, such as anti-inflammatory, cytotoxic, and neuroprotective activities, were isolated from the titled animals [5,6,7,8,9,10,11,12,13]. Their unique structures and excellent bioactivities attracted our ongoing interest to search for more bioactive secondary metabolites from the South China Sea soft corals.

Recently, the soft coral *S. nanolobata* was collected off Ximao Island, Hainan Province, China, in May 2019. Our previous chemical study on the titled animal resulted in the isolation and characterization of a series of polyoxygenated diterpenoids [14]. Further chemical investigation of the acetone extract of the same sample led to the isolation and characterization of three previously undescribed cembrane-type diterpenoids (**1**–**3**) and a new casbane-type diterpenoid (**4**) (Figure 1). Herein, we report the isolation, structure elucidation, and cytotoxic activity of these newly isolated compounds.

## 2. Results

The acetone extract of *S. nanolobata* was portioned between Et_2_O and H_2_O to afford the Et_2_O-soluble fraction, which was subjected to silica gel column chromatography to yield 12 subfractions. The subfractions were further purified using repeated silica gel, Sephadex LH-20, and reversed-phase HPLC to afford compounds **1**–**7**. The known compounds were rapidly characterized as grandilobatin D (**5**) [6], 11,12-epoxy-1*E*,3*E*,7*E*-cembratrien-15-ol (**6**) [15], and casbene (**7**) [16] by comparing the observed and reported spectroscopic data.

Compound **1** was obtained as an optically active {[*α*${]}_{D}^{20}$ −13.5 (c 0.25, CHCl_3_)} colorless oil. Its molecular formula, C_21_H_34_O_2_, was determined using the HR-EIMS ion peak at *m*/*z* 318.2554 [M]^+^ (calcd. for C_21_H_34_O_2_, Appendix A), implying 5 degrees of unsaturation. The IR spectrum showed the presence of a carbonyl group (1708 cm^−1^, Appendix A). The ^1^H NMR data (Table 1) of **1** displayed two vinyl methyls at *δ*_H_ 1.75 (3H, s) and 1.70 (3H, s), two tertiary methyls at *δ*_H_ 1.29 (3H, s) and 1.30 (3H, s), a bimodal methyl at *δ*_H_ 0.98 (3H, d, *J* = 6.9 Hz), an oxymethyl at *δ*_H_ 3.02 (3H, s), and three olefinic protons at *δ*_H_ 6.17 (1H, d, *J* = 10.7 Hz), 5.89 (1H, d, *J* = 10.7 Hz), and 5.24 (1H, m), respectively. The ^13^C NMR data together with DEPT and HSQC spectra indicated the presence of 21 carbon signals, which were classified as 6 methyls, 6 methylenes, 4 methines, and 5 quaternary carbons. The aforementioned data revealed that **1** was a cembrane-type diterpenoid and closely resembled that of co-occurring known compound, grandilobatin D (**5**) [6], with the only difference being the methoxyl group at C-15 in **1** instead of the C-15 hydroxyl group in **5**, which is in agreement with the mass data. This replacement caused the ^13^C NMR resonance of C-15 to be shifted downfield from *δ*_C_ 73.9 to 78.2 in **1**. The position of the methoxyl group at C-15 was further confirmed by the HMBC correlation from −OMe (*δ*_H_ 3.02) to C-15 (*δ*_C_ 78.2) (Figure 2). The geometries of the double bonds at Δ^1,2^, Δ^3,4^, and Δ^7,8^ were both assigned to be *E* by the observed chemical shifts (<20 ppm) of the two vinyl methyls resonance at *δ*_C_ 17.4 (C-18) and *δ*_C_ 17.6 (C-19), along with the NOESY correlations of H-2 (*δ*_H_ 6.17)/Me-18 (*δ*_H_ 1.75), H-3 (*δ*_H_ 5.89)/H_2_-14 (*δ*_H_ 2.23 and 1.97), and H-7 (*δ*_H_ 5.24)/H_2_-9 (*δ*_H_ 3.03).

The TDDFT-ECD calculation was carried out to deduce the absolute configuration of **1**, which has been proven to be a reliable structure elucidation method for the determination of the absolute configuration of natural products. In detail, torsional sampling conformational searches using MMFFs (Merck Molecular Force Field) were carried out by means of the conformational search module in the Macromodel by applying an energy window of 21 kJ/mol, which afforded 145 conformers for (12*S*)-**1**. The Boltzmann populations of the conformers were obtained based on the potential energy provided by the MMFFs, which afforded five conformers for re-optimization. The re-optimization and the following TDDFT calculations of the re-optimized geometries were all performed using Gaussian 09 at the B3LYP/6-311G(d,p) level with IEFPCM (Polarizable Continuum Model using the Integral Equation Formalism variant) solvent model for acetonitrile. Frequency analysis was performed as well to confirm that the re-optimized geometries were at the energy minima. Finally, the SpecDis 1.62 software was used to obtain the Boltzmann-averaged ECD spectra and visualize the results. Detailed comparison of the experimental ECD spectrum with those of calculated ones revealed that the Boltzmann-averaged ECD spectrum of (12*S*)-**1** displayed an identical curve compared to the experimental one (Figure 3). Consequently, the absolute configuration of **1** was determined as 12*S*.

Compound **2**, which was isolated as a colorless oil, gave the molecular formula C_21_H_34_O_2_, the same as that of **1**, on the basis of HR-EIMS ion peak at *m*/*z* 318.2558 [M]^+^ (calcd. for C_21_H_34_O_2_, 318.2553). The ^1^H and ^13^C NMR data (Table 1) of **2** were virtually identical to those of the known co-isolated compound, 11,12-epoxy-1*E*,3*E*,7*E*-cembratrien-15-ol (**6**), with the exception of a methoxyl group at C-15 in **2** instead of the C-15 hydroxyl group in **6**. The planar structure of **2** was further elucidated via ^1^H–^1^H COSY and HMBC experiments (Figure 2). The *E* geometries of the double bonds Δ^3,4^ and Δ^7,8^ in **2** were determined using the chemical shifts (<20 ppm) of the C-18 (*δ*_C_ 18.3) and C-19 (*δ*_C_ 15.1) methyl groups, which were further confirmed by the NOESY cross-peaks of H-2 (*δ*_H_ 6.14)/Me-18 (*δ*_H_ 1.74), and H-7 (*δ*_H_ 5.29)/H_2_-9 (*δ*_H_ 2.26) (Figure 2). Moreover, the NOESY correlations of H-2/Me-16 (*δ*_H_ 1.30) and H-3 (*δ*_H_ 5.82)/H_2_-14 (*δ*_H_ 2.12 and 2.03) assigned the *E* geometry of the double bond Δ^1,2^. The relative configuration of C-11 and C-12 of **2** were suggested to be the same 11*R**, 12*R** as those of **6** due to the similar NMR data and the diagnostic NOESY relationships of H-11 (*δ*_H_ 2.90)/H-13*β* (*δ*_H_ 1.33) and Me-20 (*δ*_H_ 1.25)/H-10*β* (*δ*_H_ 1.45) (Figure 2). The absolute configuration of **2** was established by the application of the TDDFT-ECD calculation method. In this case, conformational search afforded 171 conformers for (11*R*, 12*R*)-**2** and 5 conformers for re-optimization and the following TDDFT-ECD calculation. As shown in Figure 3, the Boltzmann-averaged ECD spectrum of (11*R*, 12*R*)-**2** was matched to the experimental ECD spectrum of **2**. Accordingly, the structure of **2** was elucidated as depicted in Figure 1.

Compound **3** was also obtained as a colorless oil with the molecular formula of C_21_H_34_O_2_ on the basis of HR-EIMS ion peak at *m*/*z* 318.2566 [M]^+^ (calcd. for C_21_H_34_O_2_, 318.2553). Analysis of the ^1^H and ^13^C NMR data of **3** (Table 2) revealed similarities to **2**, except for the location of the methoxyl group from the C-15 in **2** transferred to C-4 in **3**, and accompanied by the isomerization of olefins from Δ^1,2^ and Δ^3,4^ to Δ^1,15^ and Δ^2,3^, respectively. These observations were supported by the HMBC correlations from the methyl protons Me-18 (*δ*_H_ 1.31) to C-3 (*δ*_C_ 130.5), C-4 (*δ*_C_ 77.3), and C-5 (*δ*_C_ 41.7); −OMe (*δ*_H_ 3.07) to C-4 (*δ*_C_ 77.3); Me-16 (*δ*_H_ 1.81) to C-1 (*δ*_C_ 129.5) and C-15 (*δ*_C_ 131.7); and from the olefinic proton H-2 (*δ*_H_ 5.71) to C-15 (Figure 2). The large coupling constant (*J*_2,3_ = 16.3 Hz) and the ^13^C chemical shift of the methyl group Me-19 (*δ*_C_ 14.8) established the *E* geometries of the double bonds Δ^2,3^ and Δ^7,8^. Its relative configuration at C-11 and C-12 was proven to be the same 11*R** and 12*R** as those of **2** on the basis of the NOESY experiment (Figure 2). The whole relative configuration of the remaining chiral center C-4 and the distant stereochemical domain C-11/C-12 were defined using the QM-NMR calculation and DP4+ analysis [17,18]. These calculation methods utilize Bayes’s theorem to estimate the probability of the selected candidate being correct. The common stages included the generation of plausible isomers and conformational search for each isomer in the gas phase using the MMFFs as applied in the Macromodel software Schrodinger2015-2. Finally, the NMR parameters on the two possible candidate isomers (Appendix A, **3a**: 4*R**, 11*R**, and 12*R**; **3b**: 4*S**, 11*R**, and 12*R**) were calculated by the means of gauge including atomic orbitals (GIAO) method at the mPW1PW91/6-31+G(d) level of theory following the DP4+ protocols. As a result, the experimentally observed NMR data of **3** gave the best match of over 99% to the **3b** isomer (Appendix A).

With the relative configuration assigned, the following task was the determination of the absolute configuration of **3**. Similarly, TDDFT-ECD calculation method was again applied in this case to determine the absolute configuration of **3**. The conformational search of isomer (4*S*, 11*R*, 12*R*)-**3** afforded 125 conformers and obtained 5 conformers with Boltzmann populations of more than 1% for the following re-optimization and TDDFT-ECD calculations. As shown in Figure 4, the Boltzmann-averaged ECD spectrum of (4*S*, 11*R*, and 12*R*)-**3** highly matched the experimental ECD curve of **3**. In light of these evidence, the structure of compound **3** was established as depicted in Figure 1.

Compound **4** was isolated as a colorless oil, possessing the molecular formula of C_20_H_30_ by the HR-EIMS ion peak at *m*/*z* 270.2342 [M]^+^ (calcd. for C_20_H_30_, 270.2342), suggesting that **4** possessed 6 degrees of unsaturation. The ^1^H and ^13^C NMR data (Table 2) of **4** resembled those of the known co-isolated compound, casbene (**7**), with the exception of a conjugated terminal double bond in **4** instead of the vinyl methyl in **7**. This replacement caused the presence of another three olefinic protons and the absence of a methyl signal in the ^1^H NMR of **4**. The planar structure of **4** was further confirmed by the analysis of its ^1^H–^1^H COSY and HMBC correlations (Figure 2). The geometries of the double bonds Δ^3,4^ and Δ^7,8^ were assigned to be both *E* by the shielded carbon resonances of the two vinyl methyls at *δ*_C_ 15.8 (C-18) and 18.0 (C-19), along with the obvious NOESY correlations of Me-18 (*δ*_H_ 1.65)/H-2 (*δ*_H_ 1.32) and Me-19 (*δ*_H_ 1.64)/H_2_-6 (*δ*_H_ 2.16) (Figure 2). Moreover, the large coupling constants (*J*_10,11_ = 16.2 Hz) between H-10 and H-11 established the *E* geometry of the double bond Δ^10,11^. The 1,2-*cis*-configuration of C-1 and C-2 was determined by the NOE relationships of H-1/H-2/Me-16 (Figure 2) and the large ∆*δ*_C_ value (13.1 ppm) between the gem-dimethyls C-16 (*δ*_C_ 29.2) and C-17 (*δ*_C_ 16.1). Moreover, the TDDFT-ECD calculation method was also applied to determine the absolute configuration of **4**. As a result, the Boltzmann-averaged ECD spectrum of (1*S*, 2*R*)-**4** highly matched to the experimental one, while the ECD profile of enantiomer (1*R*, 2*S*)-**4** showed completely opposite curve (Figure 4). Consequently, the absolute configuration of **4** was determined to be 1*S*, 2*R*.

In the in vitro bioassay, cembrane-type diterpenoids have been well documented to display the growth inhibitory activities against various cancer cell lines [19]. Accordingly, the cytotoxic activities of all the isolated compounds **1**–**7** were evaluated in vitro against HEL (human erythroleukemia cells), H1975 (human lung adenocarcinoma cells), A549 (human non-small cell lung cancer cells), H1299 (human non-small cell lung cancer cells), and MDA-MB-231 (human breast cancer cells) by using the CCK8 and MTT methods. The dose-dependent assay was performed for the determination of IC_50_ values for the active compounds, and only compounds **1** and **5** exhibited medium cytotoxic activities against HEL cells with IC_50_ values of 37.09 and 42.37 μM, respectively, compared to that of the positive control doxorubicin (IC_50_ = 0.05 μM for HEL). In light of the above data, the primary structure–activity relationships of **1**–**7** were summarized, and the moderate potency of **1**, **5** and the inactivity of **2**–**4**, **6**, **7** suggested that the carbonyl group at C-10 seemed to have a significant impact on the cytotoxic activity against the tested cell lines. Furthermore, the structural comparison for the pair of **1** and **5** revealed that the substitutes at C-15 also contributed to the activity.

## 3. Discussion

Although this is not the first chemical investigation that we conducted on the soft coral *S. nanolobata* from the South China Sea, we still obtained some new structures from it in this study. Structurally, all the new compounds **1**–**4** shared the same cembrane or casbane-type carbon skeleton with known analogues **5**–**7**, and these molecules differed from each other mainly in different substituents or double bond positions, which suggested that they underwent a common biosynthesis pathway. In the bioassay, ketone carbonyl compounds **1** and **5** showed potential cytotoxic activities against HEL cells compared to that of inactive compounds, which provided a possible lead scaffold for further structural modifications to design novel anti-tumor drug. Further research should be conducted on the ecological roles of these bioactive secondary metabolites formed during the biosynthesis process of the soft coral.

## 4. Materials and Methods

### 4.1. The General Experimental Procedures

Optical rotations were measured on a Perkin-Elmer 241MC polarimeter (PerkinElmer, Fremont, CA, USA). IR spectra were recorded using a Nicolet 6700 spectrometer (Thermo Scientific, Waltham, MA, USA); peaks were reported in cm^–1^. The NMR spectra were measured at 300 K on Bruker DRX 400 and Avance 600 MHz NMR spectrometers (Bruker Biospin AG, Fallanden, Germany); chemical shifts were reported in parts per million (*δ*) in CDCl_3_ (*δ*_H_ reported referred to CHCl_3_ at 7.26 ppm; *δ*_C_ reported referred to CDCl_3_ at 77.16 ppm) and coupling constants (*J*) in Hz; assignments were supported by ^1^H–^1^H COSY, HSQC, HMBC, and NOESY experiments. EIMS and HR-EIMS spectra were recorded using a Finnigan-MAT-95 mass spectrometer (ThermoFisher Scientific, Waltham, USA). Semi-preparative HPLC was performed on an Agilent-1260 system equipped with a DAD G1315D detector using ODS-HG-5 (250 mm × 9.4 mm, 5 µm) by eluting with CH_3_OH–H_2_O or CH_3_CN–H_2_O system at 3 mL/min. Commercial silica gel (200−300 and 400−500 mesh; Qingdao, China) was used for column chromatography. Precoated SiO_2_ plates (HSGF-254; Yantai, China) were used for analytical TLC. Spots were detected using TLC under UV light or by heating after spraying with an anisaldehyde H_2_SO_4_ reagent. All solvents used for extraction and isolation were of analytical grade.

### 4.2. Biological Material

Specimens of titled animals were collected along the coast of Ximao Island, Hainan province, China, in May 2019, at a depth of −20 m, and were frozen immediately after collection. The high-definition photos and biological samples of the titled animals were sent to Hainan University, and the specimens were accordingly identified as *S. nanolobata* by Prof. Xiu-Bao Li. The voucher sample is available for inspection at the Shanghai Institute of Materia Medica, SIBS-CAS (No. 19-XD-12).

### 4.3. Extraction and Isolation

The frozen soft coral (856 g, dry weight after extraction) was extracted exhaustively with acetone at room temperature (3 × 5.0 L). The acetone extract (40 g) was then partitioned between Et_2_O (3 × 1.0 L) and H_2_O (3 × 1.0 L), and the Et_2_O-soluble fraction was concentrated under reduced pressure to obtain a brown residue (16.5 g). Subsequently, the residue was separated into 12 fractions (A-L) via gradient silica gel column chromatography. Fraction A (264 mg) was partially purified using semi-preparative RP-HPLC (CH_3_CN–H_2_O, 97:3, 3.0 mL/min) to yield compounds **4** (0.6 mg, *t*_R_ = 24.4 min) and **7** (4.0 mg, *t*_R_ = 29.8 min). Fraction G (583 mg) was initially chromatographed using a Sephadex LH-20 column and eluted with PE/DCM/MeOH (2:1:1), affording four subfractions (G1–G4). Purification of subfraction G3 using semi-preparative RP-HPLC (CH_3_CN–H_2_O, 60:40) yielded compounds **1** (2.5 mg, *t*_R_ = 19.4 min), **2** (39.7 mg, *t*_R_ = 20.8 min), and **3** (4.9 mg, *t*_R_ = 21.7 min). Fraction H (356 mg) was further chromatographed using a Sephadex LH-20 column and eluted with PE/DCM/MeOH (2:1:1), affording five subfractions (H1–H5). Subfraction H3 was subsequently separated via silica gel column chromatography (300–400 mesh) and eluted with PE–DCM (1:1) to give compounds **5** (21.8 mg) and **6** (3.9 mg).

#### 4.3.1. 12α-methyl-1E,3E,7E-cembratrien-10-one (**1**)

Colorless oil; [*α*${]}_{D}^{20}$ −13.5 (c 0.25 CHCl_3_); IR (KBr) *ν*_max_ = 2928, 2871, 1708, 1456, 1376, 1154, 1072 cm^−1^; UV (MeCN) *λ*_max_ 249.0 nm (log *ε* 4.65); ^1^H and ^13^C NMR data, see Table 1; HR-EIMS *m*/*z* 318.2554 [M]^+^ (calcd. for C_21_H_34_O_2_, 318.2553).

#### 4.3.2. 15-methoxyl-11,12-epoxy-1E,3E,7E-cembratrien (**2**)

Colorless oil; [*α*${]}_{D}^{20}$ −2.8 (c 0.53 CHCl_3_); IR (KBr) *ν*_max_ = 2977, 2937, 1144, 1071 cm^−1^; UV (MeCN) *λ*_max_ 192.0 nm (log *ε* 4.29); ^1^H and ^13^C NMR data, see Table 1; HR-EIMS *m*/*z* 318.2558 [M]^+^ (calcd. for C_21_H_34_O_2_, 318.2553).

#### 4.3.3. 4α-methoxyl-11,12-epoxy-1,2E,7E-cembratrien (**3**)

Colorless oil; [*α*${]}_{D}^{20}$ +9.9 (c 0.28 CHCl_3_); IR (KBr) *ν*_max_ = 2974, 2934, 1374, 1075 cm^−1^; UV (MeCN) *λ*_max_ 243.5 nm (log *ε* 3.80); ^1^H and ^13^C NMR data, see Table 2; HR-EIMS *m*/*z* 318.2566 [M]^+^ (calcd. for C_21_H_34_O_2_, 318.2553).

#### 4.3.4. 2E,7E,10E,12-casbatetraen (**4**)

Colorless oil; [*α*${]}_{D}^{20}$ −96.7 (c 0.04 CHCl_3_); IR (KBr) *ν*_max_ = 2923, 2853, 1456 cm^−1^; CD (MeCN) *λ* (∆ε) 207.5 (−3.94), 240.5 (+0.99); UV (MeCN) *λ*_max_ 203.0 nm (log *ε* 4.05); ^1^H and ^13^C NMR data, see Table 2; HR-EIMS *m*/*z* 270.2342 [M]^+^ (calcd. for C_20_H_30_, 270.2342).

### 4.4. Computational Methods

Conformational searches were carried out using the torsional sampling (MCMM) method and the MMFFs force field. Conformers above 1% of the population were re-optimized at the B3LYP/6-311G(d,p) level using the IEFPCM solvent model for acetonitrile. Subsequently, NMR calculations were performed at the PCM/mPW1PW91/6-31G(d) level, as recommended for DP4+. NMR shielding constants were calculated by using the GIAO method. Finally, the shielding constants were averaged over the Boltzmann distribution obtained for each stereoisomer and correlated with the experimental NMR data. For the resulting geometries, ECD spectra were obtained via TDDFT calculations performed with Gaussian 09 using the same functional, basis set, and solvent model as the energy optimization. At last, the Boltzmann-averaged ECD spectra were obtained using SpecDis 1.62 software.

### 4.5. Bioactivity Assays

The cytotoxicity of compounds **1**–**7** was evaluated by using the CCK8 (HEL) and MTT (H1975, MDA MB-231, A549, and H1299) methods, with doxorubicin (DOX) as the positive control. The growth inhibition of compounds on cancer cells from different tissue sources was tested using five concentration gradients. The maximum concentration of the compounds was 50 μM, diluted five times, and the cancer cells were treated with five concentration gradients for 72 h. Compounds with the highest concentration of 50 μM and an inhibition rate greater than 60% were re-screened, and the half-maximal inhibition (IC_50_) values were calculated.

## 5. Conclusions

In summary, three new cembrane-type and one new casbane-type diterpenoids were isolated and characterized from the soft coral *S. nanolobata* collected off Ximao Island, Hainan Province, China. The structures of the new compounds were established by a combination of extensive spectroscopic analysis, comparison with literature data, and DFT-based quantum chemical calculation-aided configuration analysis. In particular, the relative configuration of **3** was defined using the QM-NMR calculation and DP4+ analysis, and the absolute configurations of **1**–**4** were determined using TDDFT ECD calculations. In the in vitro bioassay, compounds **1** and **5** exhibited moderate cytotoxic activities against HEL cells, with IC_50_ values of 37.09 and 42.37 μM, respectively. The discovery of these new bioactive secondary metabolites once again proved the chemical diversity of the soft coral *S. nanolobata*.

## Figures and Tables

**Figure 1 molecules-28-06892-f001:**
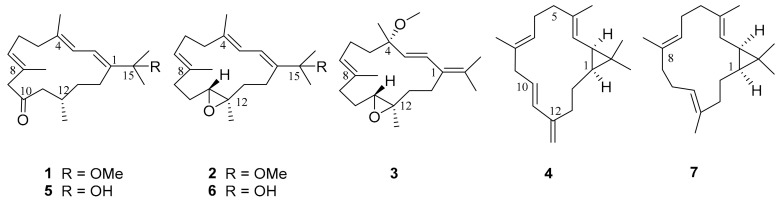
The chemical structures of compounds **1**–**7**.

**Figure 2 molecules-28-06892-f002:**
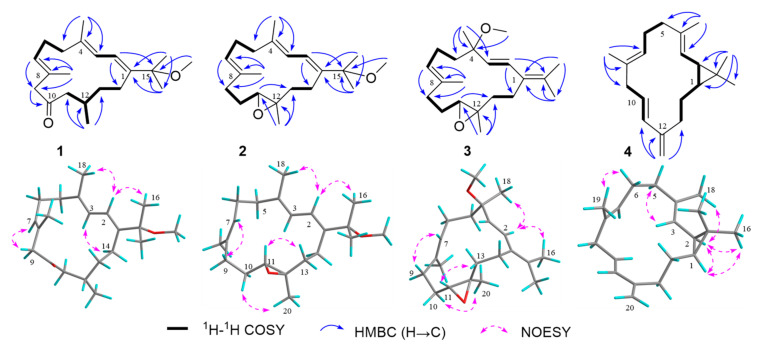
The ^1^H–^1^H COSY, key HMBC, and NOESY correlations of compounds **1**–**4**.

**Figure 3 molecules-28-06892-f003:**
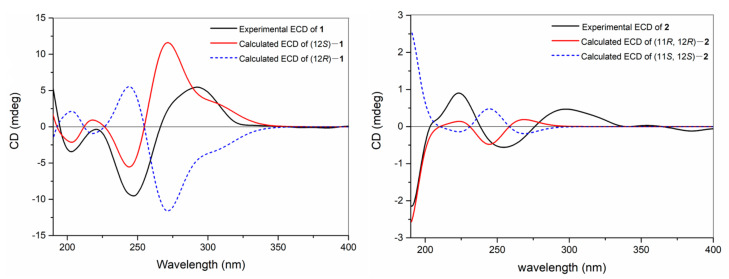
Experimental and calculated ECD spectra of **1** and **2**.

**Figure 4 molecules-28-06892-f004:**
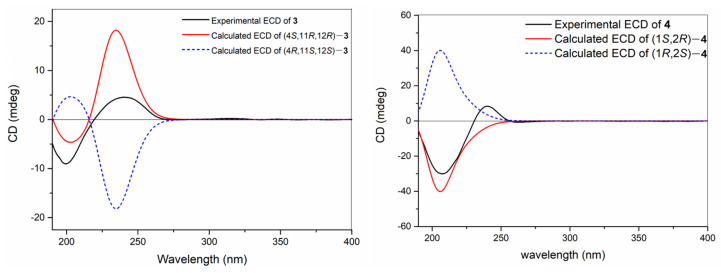
Experimental and calculated ECD spectra of **3** and **4**.

**Table 1 molecules-28-06892-t001:** The ^1^H and ^13^C NMR data of **1** and **2** in CDCl_3_ ^a^.

No.	1		2	
	*δ*_H_, Mult (*J* in Hz)	*δ*_C_, Mult	*δ*_H_, Mult (*J* in Hz)	*δ*_C_, Mult
1	-	143.2, s	-	143.0, s
2	6.17, d (10.7)	122.1, d	6.14, d (10.0)	122.3, d
3	5.82, d (10.7)	121.8, d	5.82, d (10.0)	120.3, d
4	-	137.3, s	-	138.8, s
5	2.18, m	39.0, t	2.17, m	38.2, t
	2.18, m		2.17, m	
6	2.23, m	25.5, t	2.27, m	25.1, t
	2.23, m		2.17, m	
7	5.24, m	128.9, d	5.28, m	127.2, d
8	-	129.4, s	-	133.6, s
9	3.03, m	53.1, t	2.26, m	37.0, t
	3.03, m		2.11, m	
10	-	209.9, s	2.01, m	24.5, t
			1.45, m	
11	2.55, dd (14.6, 8.7)	50.9, t	2.90, dd (9.2, 3.5)	61.3, t
	2.19, m			
12	2.06, m	30.2, d	-	61.4, d
13	1.43, m	37.4, t	2.08, m	38.8, t
	1.27, m		1.33, m	
14	2.23, m	24.6, t	2.12, m	23.1, t
	1.97, m		2.03, m	
15	-	78.2, s	-	78.0, s
16	1.29, s	27.0, q	1.29, s	26.4, q
17	1.30, s	25.2, q	1.29, s	25.8, q
18	1.75, s	17.4, q	1.74, s	18.3, q
19	1.70, s	17.6, q	1.66, s	15.1, q
20	0.98, d (6.9)	20.4, q	1.25, s	17.4, q
−OMe	3.02, s	50.4, q	3.02, s	50.4, q

^a 1^H NMR at 600 MHz, values are reported in ppm referenced to CHCl_3_ (*δ*_H_ 7.26). ^13^C NMR at 150 MHz, values are reported in ppm referenced to CDCl_3_ (*δ*_C_ 77.16). Assignments were aided by HSQC and HMBC experiments.

**Table 2 molecules-28-06892-t002:** The ^1^H and ^13^C NMR data of **3** and **4** in CDCl_3_ ^a^.

No.	3		4	
	*δ*_H_, Mult (*J* in Hz)	*δ*_C_, Mult	*δ*_H_, Mult (*J* in Hz)	*δ*_C_, Mult
1	-	129.5, s	0.62, dt (8.2, 2.6)	29.5, d
2	6.48, d (16.3)	127.4, d	1.32, m	26.0, d
3	5.71, d (16.3)	130.5, d	4.83, d (8.2)	122.9, d
4	-	77.3, s	-	134.8, s
5	1.92, m	41.7, t	2.22, dd (12.1, 4.2)	39.6, t
	1.60, m		2.06, dd (12.1, 4.9)	
6	2.64, m	23.0, t	2.16, m	23.8, t
	1.95, m		2.16, m	
7	5.34, br d (7.7)	128.6, d	5.14, t (5.7)	124.2, d
8	-	132.5, s	-	134.2, s
9	2.33, d (13.0)	36.9, t	2.73, dd (16.4, 5.0)	40.8, t
	2.10, dd (13.0, 3.1)		2.62, dd (16.4, 9.1)	
10	2.18, dt (12.9, 3.0)	24.4, t	5.67, ddd (16.2, 9.1, 5.0)	130.5, d
	1.32, m			
11	2.79, dd (10.8, 2.6)	62.6, d	5.94, d (16.2)	130.8, d
12	-	61.6, s	-	147.5, s
13	2.01, m	37.6, t	2.31, m	34.3, t
	1.02, m		2.31, m	
14	2.46, m	26.4, t	1.49, m	25.5, t
	2.04, m		1.38, m	
15	-	131.7, s	-	20.1, s
16	1.81, s	21.5, q	1.07, s	29.2, q
17	1.81, s	20.4, q	0.93, s	16.1, q
18	1.31, s	23.3, q	1.65, s	15.8, q
19	1.70, s	14.8, q	1.64, s	18.0, q
20	1.30, s	16.3, q	4.87, s	113.0, t
			4.82, s	
−OMe	3.07, s	50.3, q		

^a 1^H NMR at 600 MHz, values are reported in ppm referenced to CHCl_3_ (*δ*_H_ 7.26). ^13^C NMR at 150 MHz, values are reported in ppm referenced to CDCl_3_ (*δ*_C_ 77.16). Assignments were aided by HSQC and HMBC experiments.

## Data Availability

Data are available in Electronic Appendix A (ESI).

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
