# Peer review of "Four New Diterpenoids from the South China Sea Soft Coral Sinularia nanolobata and DFT-Based Structure Elucidation"

_molecules, 2023, doi:10.3390/molecules28196892_

Round 1

Reviewer 1 Report

In the paper “Four new diterpenoids from the South China Sea soft coral Sinularia nanolobata and DFT-based structure elucidation” the authors isolated and structurally characterized several new compounds with potential biological activity. The manuscript is well written with flawless English. The conclusions are supported by experimental results and DFT calculations. I have just two comments. I think that the paragraph about the experimental setup of DFT calculations of ECD spectra should be transferred from the Supplement to the main part of the paper. It would help to improve the logic of the paper. The second comment deals with a lack of any description of the evaluation of NMR spectra by DP4+. Otherwise, by my opinion, the manuscript can be published without any other, changes.

Reviewer 2 Report

The manuscript submitted by X. Luo and Yue-Wei Guop et al. report the isolation and characterization of four new diterenoids 1-4 from a pool of remarkably diverse metabolites of the soft coral Sinularia nanolobata from the South China Sea. The authors skilfully used the combination of experimental NMR and ECD spectra together with GIAO (for 3) and TDDFT calculations to assign structure and absolute configuration of the isolated compounds and demonstrated their moderate cytotoxicity.

In my opinion, it is a virtually flawless well-written work which is definitely worth of publishing in the special issue “Advances in Computer Assisted Structure Elucidation” of the journal “Molecules”. Minor English editing might be required, for example line 37 “which exhibiting” should be corrected to “which exhibited”.

I wish the authors had provided more detailed experimental data on the isolation of the compounds, indicating the mass of soft coral sample used for extraction, fraction volumes and RP-HPLC profiles, to give the reader an idea of the diterpenoid content in the samples studied against the background of the other metabolites.

Another point is not a requirement to be fulfilled in the course of the revision, but rather a recommendation for further work. A few years ago Bannwarth and Grimme introduced a family of simplified TDA and TDDFT approximations (Comput. Theor. Chem., 2014, 1040–1041, 45–53; Chirality, 2016, 28, 365–369; J. Am. Chem. Soc., 2017, 139, 11682–11685) to predict the ECD spectra of biomolecules. I think that the diterpenoids studied by the authors could be good models to test these methods and prove their validity for a wider range of objects.

Minor English editing might be required
